# Process evaluation of a hybrid effectiveness-implementation, pragmatic, cluster randomised controlled trial (IMPULSE) to improve psychosocial treatment of patients with psychotic-spectrum disorders in Southeast Europe

Tamara Pemovska[1,2], Nikolina Jovanović[2*], Tamara Radojičić[3,4], Silvana Markovska Simoska[5], Fjolla Ramadani[6], Sanja Andrić Petrović[7,8,9], Emina Karamehić[10,11], Biljana Blazhevska Stoilkovska[12], Jon Konjufca[6,13], Stefan Jerotić[7,9], Alma Džubur Kulenović[11ᵒ], Lidija Injac Stevović[3ᵒ], Jill J. Francis[14]

1 Centre for Evidence and Implementation, London, United Kingdom, 2 Centre for Psychiatry and Mental Health, Wolfson Institute of Population Health, Queen Mary University of London, London, United Kingdom, 3 Psychiatric Clinic, Clinical Centre of Montenegro, Podgorica, Montenegro, 4 ATLANTES Global Observatory of Palliative Care, Institute for Culture and Society, University of Navarra, Pamplona, Spain, 5 Macedonian Academy of Sciences and Arts, Skopje, North Macedonia, 6 Department of Psychology, University of Prishtina, Prishtina, Kosovo United Nations Resolution, 7 Faculty of Medicine, University of Belgrade, Belgrade, Serbia, 8 Institute of Mental Health, Belgrade, Serbia, 9 Clinic for Psychiatry, University Clinical Centre of Serbia, Belgrade, Serbia, 10 Public Institution Department for Health Care of Women and Maternity of Sarajevo Canton, Sarajevo, Bosnia and Herzegovina, 11 Department of Psychiatry, Clinical Centre of the University of Sarajevo, Sarajevo, Bosnia and Herzegovina, 12 Department of Psychology, Faculty of Philosophy, Ss. Cyril and Methodius University in Skopje, Skopje, North Macedonia, 13 Department of Psychology, University of Basel, Basel, Switzerland, 14 School of Health Sciences, University of Melbourne, Australia

ᵒ These authors contributed equally to this work.
* n.jovanovic@qmul.ac.uk

## Abstract

### Background

The IMPULSE trial investigated the effectiveness and implementation of a digital psychosocial intervention (DIALOG+) for people with psychosis in five Southeast European countries. DIALOG+ significantly improved patients' quality of life after four treatment sessions. The process evaluation reported here aimed to assess contextual influences on intervention delivery during the trial, to explain the trial findings and generate hypotheses about mechanisms of action by exploring acceptability from the perspectives of clinicians who delivered it and trial participants who received it, and fidelity (was the intervention delivered and received as planned?).

### Method

A mixed-methods process evaluation was conducted in accordance with the published protocol, guided by theoretical frameworks and the Medical Research

**Data availability statement:** The dataset (which includes individual transcripts) used and analysed during the current study is not publicly available due to confidentiality policies. The authors do not have the ethical approval to make data immediately available to the public. The project did not explicitly seek the informed consent of participants for the future public release of their data. Data are from the IMPULSE study. data access requests may be directed to Queen Mary University of London, Wolfson Institute of Population Health; email: wiph-admin@qmul.ac.uk.

**Funding:** The authors disclosed receipt of the following financial support for the research, authorship, and publication of this article: This study was funded as part of the IMPULSE project under the European Union's Horizon 2020 research and innovation programme [grant number 779334]. The IMPULSE project has received funding through the "Global Alliance for Chronic Diseases (GACD) prevention and management of mental disorders" (SCI-HCO-07-2017) funding call. This paper presents independent research funded by the European Union – the funding role had no role in study design, data collection and analysis, decision to publish, or preparation of the manuscript.

**Competing interests:** The authors declare that they have no competing interests.

Council's guidance for complex interventions. To explore the role of context, data were analysed about the participating services, policy documents, and from focus groups with key stakeholders. Semi-structured interviews with clinicians and patients were conducted to explore acceptability. Process data (format and content of sessions) were analysed to assess intervention fidelity. Data analysis included descriptive methods, framework and content analysis, and triangulation.

## Results

Several attributes of context related to health services, including resource limitations, funding priorities, reliance on paper records and lack of community support, potentially negatively impacted DIALOG+ acceptability, fidelity and outcomes. Contextual enablers were also identified, including an appetite for change among key stakeholders that can help overcome contextual barriers. Acceptability of the psychosocial intervention was moderate to high and fidelity was high.

## Conclusions

Intervention acceptability is likely to have played a key role in ensuring high fidelity, which in turn likely contributed to the intervention's positive impact on patients' quality of life. The high fidelity confirms that the IMPULSE trial findings provide a valid assessment of the intervention as designed. While the identified contextual barriers appear not to have impaired intervention fidelity, acceptability and outcomes, they could pose challenges to the long-term sustainability of the intervention.

## Trial registration

Retrospectively registered on 29 March 2021, ISRCTN11913964

## Introduction

International treatment guidelines for psychotic-spectrum disorders (PSD) suggest a combined-therapy approach which includes antipsychotic medication and non-pharmacological interventions such as educational, psychotherapeutic, social, and physical interventions. In low- and middle-income countries (LMICs) most of the available mental health funds are spent on inpatient care, while community-based services are scarce [1]. Additionally, non-pharmacological treatment approaches that can meaningfully improve people's quality of life are rarely available and the human resources to provide these interventions are inadequate [2].

The IMPULSE project received funding to improve community-based, psychosocial care of individuals with psychotic disorders in five Southeast European LMICs: Bosnia and Herzegovina, Kosovo United Nations (UN) Resolution, Montenegro, North Macedonia, and Serbia. The IMPULSE cluster-randomized control trial (cRCT) sought to evaluate the effectiveness, cost-effectiveness and implementation of a psychosocial intervention, called DIALOG+, for patients with psychotic disorders

(ISRCTN11913964) [3]. The core elements of DIALOG+ are consistent with the principles of solution-focused therapy, such as strengthening and utilising the resources of patients and offering guidance for individual problem-solving, and include structured and repeated digital tablet-assisted assessments of patients' satisfaction with multiple areas of their life and treatment, agreement of actions for the patient to complete between intervention sessions, and patient-centred communication characterized by encouragement and social reinforcement. DIALOG+ was previously shown to be effective in improving the quality of life and reducing clinical symptoms in long-term patients with PSD in community healthcare settings in the United Kingdom [4]. A process evaluation of the UK trial suggested that the intervention improves patients' quality of life by providing a positive change in patients' specific areas of concern, comprehensive and solution-focused structure to the routine clinical meetings, opportunity for self-reflection and therapeutic self-expression, and empowerment [5]. The IMPULSE cRCT, reported separately, showed that DIALOG+ significantly improved patients' quality of life after just four treatment sessions [6] and the economic evaluation suggested that DIALOG+ required relatively low costs [7].

Process evaluations nested within randomized controlled trials (RCTs) are crucial to the evaluation of complex interventions because they show how an intervention and its trial outcomes can be replicated in practice [8]. Robust process evaluations of RCTs investigating psychosocial interventions with individuals with psychotic disorders have been limited, particularly in low resource settings, such as Southeast Europe. The present process evaluation study aimed to assess the role of context in implementing the intervention across various mental healthcare settings. It also sought to interpret the IMPULSE cRCT findings and generate hypotheses about the intervention's mechanisms of action by exploring intervention acceptability as reported by clinicians and patients in the intervention arm and by examining intervention fidelity (was the intervention delivered and received as planned?).

## Materials and methods

The protocol of this process evaluation, explaining the methodology in detail, has been published previously [9].

### Study design

This process evaluation of an effectiveness-implementation cRCT of DIALOG+ in outpatients with PSD in five Southeast European countries (Bosnia and Herzegovina, Kosovo UN Resolution, North Macedonia, Montenegro and Serbia) adopted a mixed-methods study design and involved data collection and analysis using qualitative and quantitative methods. It was developed according to the UK Medical Research Council's guidance for process evaluations of complex interventions [8]. The CONSORT diagram of the cRCT is reported elsewhere [6].

### Setting

Eleven mental health services (including hospital-based, outpatient and community-based services) participated in the trial across the five countries. The recruitment and geographical location of the participating services, and their standard outpatient mental health care, are described in the IMPULSE cRCT protocol [3].

### Intervention and IMPULSE cRCT

DIALOG+ is a psychosocial intervention delivered through an app on a tablet computer. It begins by asking patients to self-assess their satisfaction with eight life areas and three treatment areas, on a 7-point scale, called DIALOG scale, with higher scores indicating higher satisfaction. Next, the mental health professional and patient together review ratings across all areas and patients are encouraged to choose an area to discuss in more detail during which the mental health professional uses a four-step, solution-focused approach (1-Understanding, 2-Looking forward, 3-Exploring options, 4-Agreeing on actions), focused on strengthening and utilising the resources of individuals, rather than on deficits. The DIALOG+ session finishes by agreeing clear actions that the patient, clinicians and/or another person from the patient's

circle should aim to achieve before the next session. The interaction between the clinician and the patient during the DIALOG+ session is characterized by encouragement and social reinforcement, patient-centred communication, and high patient involvement [10].

The IMPULSE cRCT was designed as a two-arm study – DIALOG+ intervention and treatment as usual as a control arm. Recruitment of participants took place between 1st of February and 1st of September 2019. Randomisation was conducted at the clinician-level to prevent clinicians having to treat patients in both arms, thereby minimizing contamination between trial arms. Thus, clinicians were randomised 1:1 to deliver either the DIALOG+ intervention or treatment as usual (control). The delivery of the intervention was guided by an implementation strategy that was primarily informed by focus group data from the DIALOG+ pilot study, mapped onto the Theoretical Domains Framework of behaviour change to identify facilitators and barriers to delivering and engaging with the intervention, with suitable behaviour change techniques incorporated to address the identified potential barriers to implementation [6,11,12]. DIALOG+ was implemented by members of the local research teams providing a training programme to clinicians in the intervention arm, which included face-to-face core training before starting to deliver the intervention, top-up training after delivery of the first and fourth DIALOG+ session, with additional supervision available as needed. Psychiatrists, psychologists and mental health nurses were trained to deliver all components of the intervention to patients with PSD. The clinicians were encouraged to deliver the intervention during face-to-face routine clinical meetings, six times during the twelve-month study period. Patients were provided with booklets where they could note the agreed actions. Information about recruitment, randomisation, trial design and outcome measures is reported in the IMPULSE cRCT protocol [3].

## Data collection

### Role of context

**Quantitative data.** Data related to the capacity of the eleven participating mental health services, such as number of patients seen, number of clinicians in the service, frequency of routine meetings etc., were collected during initial site visits prior to trial commencement, to investigate the settings in which DIALOG+ implementation was planned during the trial [3,9]. A summary table of the collected data was published in one of the IMPULSE project deliverable reports [13], which was used to explore this objective of the process evaluation.

**Qualitative data.** Prior to the start of the trial, views on the perceived barriers and facilitators to engaging with DIALOG+ were collected through 32 focus groups with 174 participants (patients, clinicians, policymakers and carers) from the five participating countries. Before each focus group, the participants were familiarized with DIALOG+ through a PowerPoint presentation containing a standardized description of the intervention. Additional qualitative data from mental health policy documents from the five participating countries were collected prior to the trial. The findings from these data have been published as IMPULSE project deliverable reports [14,15], with the focus group study also published as a scientific article [16]. These findings were further used to explore the settings in which DIALOG+ implementation was planned during the trial and contributed to the strategy for the intervention implementation. Qualitative data, such as type of service and type of treatment received in the participating mental health services was also documented during the initial site visits [13].

### Intervention acceptability

**Qualitative data.** Topic guides were developed iteratively by a multidisciplinary team for exploring the experience of using DIALOG+ and its perceived sustainability (reported elsewhere [17]) for clinicians and patients from the intervention arm (See S1 File).

All recruited patients (n = 236) and clinicians (n = 41) in the intervention arm of the trial were approached about participating in the end-of-trial qualitative study. Researchers were instructed to identify 8–10 patients per research site, from those who consented to participate in the qualitative study, to interview while seeking variability in age, gender, level of

engagement with DIALOG+, diagnosis and cluster allocation (no more than 2 patients from the same clinician cluster). Thus the sampling was purposive and began in February 2020. Both patients and clinicians were asked to consent to this study at the beginning of the IMPULSE trial. Participants received a compensation of €25 for their time.

An intensive training in qualitative research related to data collection and analysis was provided to the interviewers in the research team to ensure a unanimous approach when interviewing the clinicians and patients. The interviews with clinicians took place after they completed their final DIALOG+ session as part of the trial, whereas the interviews with patients were scheduled as soon as they completed their final follow-up assessment of the clinical trial. Interviews took place between May and August 2020.

Semi-structured interviews were audio-taped and transcribed verbatim. Any identifying information was removed to maintain anonymity and confidentiality. Approximately 20% of the transcripts from each participant group were translated into English to enable collaboration, with most transcripts remaining in their original languages to minimise the risk of mistranslation and loss of shades of meaning. In the results presented below, interviewees whose quotes are included are referred to with identification numbers created using the following strategy: letter **P** or **C** for patient or clinician, respectively; site where the interview took place **BIH** (for Bosnia and Herzegovina), **KOS** (for Kosovo UN Resolution), **MAC** (for North Macedonia), **MON** (for Montenegro), **SER** (for Serbia); consecutive number of interviewees; letter **R** for researcher/ interviewer.

### Intervention fidelity

Intervention fidelity was assessed according to the constructs proposed by Bellg et al. [18] and Carroll et al. [19]: training of clinicians, delivery of intervention, receipt of intervention and enactment of plans agreed during the intervention sessions, and intervention differentiation (extent to which the active intervention differed from the control condition).

**Quantitative data.** From February 2019 to December 2020, data on the training sessions and quantitative process outcomes (number of sessions attended, their duration and content for, e.g., type of areas chosen) were collected by local researchers in the five participating countries.

After clinicians completed their first DIALOG+ session with all their allocated patients, local researchers recorded if preparation and training for intervention delivery had been completed and if clinicians experienced any problems delivering the intervention. These data were collected for all clinicians in the intervention arm from February to June 2019.

During June 2019 – July 2020, audio recordings were obtained from one session per clinician (in intervention and control arms) who, together with the patient, agreed to be audio-recorded. The audio recordings, obtained from the third to sixth sessions to allow for clinicians to get accustomed to the intervention, were analysed using the DIALOG+ Adherence Scale (DAS) [20] that measured clinicians' adherence to the intervention manual. A short quantitative survey was used to measure patients' receipt and enactment of activities that had been agreed in the previous session.

**Qualitative data.** After clinicians completed their first DIALOG+ sessions with all their allocated patients, local researchers asked them to report how they resolved any issues occurring during intervention delivery. Clinicians' answers in the form of qualitative data were collected between January 2019 and December 2020 from all clinicians in the intervention arm. Additionally, after each intervention session, local researchers in the five participating countries collected qualitative data of the action items agreed during the sessions. Further qualitative data regarding the perceived usefulness of the training materials, intervention receipt and enactment of the agreed plans during the intervention was collected during the end-of-trial semi-structured interviews, described in detail in the next section.

### Deviations from published protocol

Originally the research team planned to conduct individual face-to-face semi-structured interviews with patients and face-to-face focus groups with clinicians. However, in the context of the COVID-19 pandemic, it was no longer possible to conduct qualitative data collection face-to-face, thus the procedure was updated to remote qualitative data collection. Only semi-structured individual interviews were conducted due to the complexity of conducting focus groups remotely.

## Data analysis

### Role of context

The analysis followed the procedure described in the process evaluation protocol [9]. Findings from all three sources of data (site visits reports, policy analysis and pre-trial focus group analysis) were coded onto a framework of the attributes and features of context developed by Squires and colleagues [21]. The attributes with their definitions are included in Table 1. After familiarizing themselves with the contextual attributes by Squires et al. [21], two researchers (TP and SM) conducted the coding by categorizing the results from the three data sources and summarizing them into the framework of contextual attributes using Microsoft Excel software. For each contextual attribute the data was classified based on interpretation as a barrier or a facilitator. Triangulation of the findings from the three different data sources was conducted by two researchers (TP and SM) following the triangulation protocol described by Farmer et al. [22]. Frequent team meetings were held during the analysis process to discuss any concerns and coding inconsistencies.

### Intervention acceptability

We utilized the Theoretical Framework of Acceptability (TFA) [23] to investigate intervention acceptability. We defined experienced acceptability based on Sekhon et al. [23]: "a multi-faceted construct that reflects the extent to which people delivering or receiving the healthcare intervention consider it to be appropriate, based on experiential cognitive and emotional responses to the intervention". The seven component constructs of acceptability with their definitions are presented in Table 2.

The end-of-trial interviews were analysed using framework analysis [24], following the steps of the analysis process described in the published protocol [9]. Data was analysed by a data analysis coordinating team including at least one researcher from each of the participating countries (TP, TR, SA, FR, JK, SM, BB, EK). The analysis team

**Table 1. Contextual attributes and their definitions (Adapted from Squires et al. [21] to relate to the study's setting of interest).**

| Contextual attribute | Definition |
| --- | --- |
| 1) Resource Access | Access to resources in a mental health care setting, such as time, staff, treatment programmes, documentation. |
| 2) Work Structure | The arrangement of tasks and resources of mental health clinicians (e.g., continuity of care, standardization of care, timeframe). |
| 3) Financial | Costs, funding systems and financial incentives in a mental health care setting. |
| 4) Patient Characteristics | Attributes of individuals under treatment for psychosis. |
| 5) Facility Characteristics | Attributes of mental health care facilities (e.g., type of facility, geography, atmosphere, volume of patients). |
| 6) Professional Role | A set of expectations related to a mental health clinical role (e.g., clinical skill set, training, job autonomy). |
| 7) Culture | Inherited ideas, beliefs, values, and attitudes of the stakeholder groups in the mental healthcare system (e.g., organisational culture). |
| 8) Health Care Professional Characteristics | Attributes of mental health clinicians (e.g., experience) |
| 9) Collaboration | Interactions between stakeholder groups in mental healthcare system. |
| 10) Evaluation | Systematic collection of information to evaluate mental health care and inform its future development. |
| 11) System Features | Characteristics of a group of related parts that work together for mental health services to run effectively. |
| 12) Societal Influences | Attitudes about mental health care. |
| 13) Leadership | The direction of management of mental health services |
| 14) Regulatory or Legislative Standards | Binding standards that are established by structures outside of the control of mental health organizations. |

**Table 2.** The component constructs of the Theoretical Framework of Acceptability (TFA) and their definitions (Adapted from [23], http://creativecommons.org/licenses/by/4.0/, to relate to the DIALOG+ intervention).

| TFA component construct | Definition |
| --- | --- |
| Affective attitude | How an individual feels about the DIALOG+ intervention. |
| Burden | The perceived amount of effort that is required to participate in DIALOG+. |
| Ethicality | The extent to which the DIALOG+ intervention has a good fit with an individual's value system. |
| Intervention coherence | The extent to which the participant understands DIALOG+ and how it works. |
| Opportunity cost | The extent to which benefits, profits or values must be given up to engage in the DIALOG+ intervention. |
| Perceived effectiveness | The extent to which the DIALOG+ intervention is perceived as likely to achieve its purpose. |
| Self-efficacy | The participant's confidence that they can perform the behaviour(s) required to participate in DIALOG+. |

began by reading English translations of the collected transcripts (17 transcripts were translated) and meeting to exchange our initial analytical impressions from the data. This was followed by individual 'open coding' and development of coding lists for patients and clinicians for each TFA component construct. Several discussions were held to compare individual coding lists and after multiple revisions, consensus was reached on a preliminary analytical framework. It was decided to use the same analytical framework for both clinicians' and patients' transcripts since the topic guide questions for both participant groups were largely the same and there was significant overlap between the emerging codes. The analysis team piloted the analytical framework on two translated transcripts and on ten original transcripts from all countries. Further refinements of the analytical framework were made based on the piloting process, such as adding additional distinct categories/codes, merging existing codes and clarifying the scope of the codes and categories. After, researchers organized their transcripts in local languages into the framework codes/categories using the RQDA software. The team met regularly to discuss any uncertainties during this process and further modifications of the analytical framework. The indexing stage was followed by charting which was completed using Microsoft Excel. The chart, organized based on the analytical framework, contained a short summary of the data from each participant relating to the particular code and an illustrative quote in English. Next, two researchers from the team (TP, TR) reviewed half of the chart and interpreted a set of themes in light of the research question of interest. These preliminary themes were presented to the rest of the team to check their credibility and clarity, and were modified accordingly. The remaining part of the chart was analysed by the same two researchers, the themes were further revised and presented to the wider team, after which the interpreted themes were finalized and categorized according to the TFA constructs: 'Affective attitude', 'Burden', 'Opportunity-cost', 'Ethicality', 'Self-efficacy', clinicians' 'Intervention Coherence' and 'Effectiveness'.

### Intervention fidelity

Intervention fidelity explored training of clinicians, delivery of intervention, receipt of intervention and enactment of plans agreed during the intervention sessions, and intervention differentiation, as proposed by Bellg et al. [18] and Carroll et al. [19].

Quantitative data was analysed using descriptive statistics, such as frequencies, means and standard deviations (SD). Qualitative survey data were analysed by two to three researchers (TP, TR, FR) using conventional content analysis following the methodology described by Hsieh & Shannon [25]. We began by reading through the data, followed by open coding and deciding on preliminary lists of codes and key categories in relation to our research question. During the coding process, new codes were added to the list, and some codes were combined, split or renamed. Regular meetings were held to discuss any uncertainties or proposed changes to the coding lists. The coded data was reviewed several times throughout the analysis process until no further modifications of the coding list were needed. Key categories are reported below and, where appropriate, their frequencies. Qualitative data from the end-of-trial interviews regarding

receipt, enactment and usefulness of training materials was analysed using framework analysis following the methodology described by Ritchie & Spencer [24] detailed in the following section.

### Ethical considerations

All research activities were conducted according to the ethical standards of the institutional and/or national research committee and consistent with the 1975 Helsinki declaration and its later amendments. All procedures were approved by the relevant ethics committees ahead of the study beginning: United Kingdom (Queen Mary University of London QMREC2204a, 16/10/2018), Bosnia and Herzegovina (Klinicki Centar Univerziteta u Sarajevu—Eticki Komitet 03-02-4216, Eticki komitet JU Psihijatriska bolnica Kantona Sarajevo & JU Zavod za bolesti ovisnosti Kantona Sarajevo 02.8–408/19), Serbia (Eticka komisija Medicinskog fakulteta u Beogradu 2650/XII-20 and Eticka komisija Specijalne bolnice 'Dr Slavoljub Bakalovic' Vrsac 01–36/1), Kosovo UN Resolution (Hospital and University Clinical Service of Kosovo —Ethics Committee 2019–85), Republic of North Macedonia (Eticka Komisija za istrazuvanje na luge, Medicinski Fakultet pri UKIM vo Skopje 03–24219), and Montenegro (Javna Zdravstvena Ustanova Klinicki Centar Crne Gore—Eticki komitet 03/01–29304/1, ZU Specijalna Bolnica za Psihijatriju "Dobrota" Kotor—Eticki komitet, Eticki Komitet JZU Dom Zdravlja "DR Nika Labovic" Berane 01–47).

The local researchers provided detailed study information to potential participants and obtained written informed consent before data collection. Researchers explained the study to the participants and provided all relevant information, such as the study's risks and benefits as well as information about their confidentiality and privacy. Participants continued their routine treatment throughout the duration of the study and were made aware that they could withdraw at any point if they wished to do so. The researchers were instructed to end the research activity and contact a clinician as soon as any participating patient appeared highly distressed or upset.

### Study registration

This study was retrospectively registered while recruitment was underway at www.isrctn.com on 29 March 2019 (ISRCTN11913964).

## Results

The results are organized per study objective and a summary of all findings is provided at the end of this section. The participants of the IMPULSE cRCT are described elsewhere [6]. In total, 81 clinicians (41 intervention, 40 control arm) were randomized. They treated a total of 468 patients (236 intervention, 232 control arm). On average, 5.3 DIALOG+ sessions (SD = 1.8) were delivered over 12 months. At 6 months, patients in the intervention arm reported improved quality of life (p = 0.03), with no other significant outcome differences. Twelve-month data were not interpretable due to COVID-19 disruptions.

### Role of context

We identified 11 out of the 14 attributes of context in the framework developed by Squires and colleagues [21]. Findings from three sources of data (site visits reports, policy analysis and pre-trial focus group analysis) were compared in the triangulation analysis by labelling each of the contextual attributes in terms of Agreement (between data sources), Partial agreement, Silence or Dissonance. Detailed findings from the triangulation step of the analysis are presented in S1 Table.

Six attributes of context that potentially negatively impacted on DIALOG+ acceptability, fidelity and outcomes were identified. The majority relate to the health services and systems: limited access to resources such as time, staff, space and equipment; funding prioritization of quantity of care with lack of funds for specialized therapies and technological equipment; reliance on paper records; dominant hospital-based mental health care treating a large volume of patients with psychiatrists being the main service provider; lack of community mental health centres (CMHCs); and a lack of citizens'

organizations that support patients, their families and health care services to run effectively. One attribute of context was associated with clinicians: moderate expectations of mental health clinicians to possess skills in psychotherapy and technology. Another contextual attribute related to patient characteristics: perception of low cognitive capacity among the patient population with PSD when they are unwell.

Eleven attributes of context were identified as potential facilitators to DIALOG+ acceptability, fidelity and outcomes. Most of them relate to the mental health services and systems, such as availability of necessary resources, some presence of CMHCs, universal health coverage for psychological interventions, presence of electronic medical records, an overall appetite for improving existing outpatient mental health care among key stakeholders (better tracking of patients' progress, prolonged outpatient meetings, inclusion of nurses in psychosocial intervention delivery, and a shift from a medical model of care towards patient-centred care focused on patients' quality of life), some existence of citizens' organizations focused on mental health, established laws for protection of people with mental illness, and a culture that values inclusion of carers in patients' care. Those associated with clinicians included: experience with psychosocial interventions and technology in their provision of routine care, motivation to reinforce community services and a collaborative network between existing mental health staff. Additionally, we identified patients' ability to be actively involved in their treatment and use technology as potential contextual facilitators.

The comparison of the three data sources identified considerable 'silence' in relation to identified contextual attributes, i.e., each data source provided unique contributions to our understanding of the context where DIALOG+ was implemented during the cRCT. The findings were in partial agreement regarding the number of staff accessible in the mental health facilities for the contextual attribute, access to resources. Additionally, the results from all three data sources were in agreement regarding the type of treatments provided in the participating mental health facilities. These largely included pharmacotherapy and, to a lesser extent, various psychological interventions. Moreover, the findings from the site visits reports and policy analysis were in agreement regarding the maintenance of patient records for the contextual attribute about features of the health system.

**Experienced intervention acceptability**

**Participants.** The total number of patients interviewed was 40; 17% of the total sample of patients in the intervention arm (n = 236). All intervention clinicians who consented to participate were interviewed (n = 35/41, 85%).

Table 3 shows the characteristics of interviewed patients and clinicians, interviewers and methods used to conduct interviews. There was an even split of gender among the patients (male n = 21, 52.5%), whereas the majority of interviewed clinicians were female (n = 28, 80%). Patients were on average 43.20 years old (SD = 10.04, range 22–67), while clinicians had an average age of 46.51 (SD = 8.04, range 26–65). The average years of clinical experience for the clinician participants was 16.69 years (SD = 9.28, range 2–39). The majority of the patient participants (n = 32, 80%) had diagnosis of schizophrenic and related disorders (ICD-10 F20-29), whereas the remaining patients had diagnosis of mood affective disorders (ICD-10 F31). Further details of these characteristics by site are available in S2 Table. Moreover, 97.5% of the patients (n = 39) attended all 6 intervention sessions, and one patient attended 5 sessions. Additionally, 82.5% (n = 33) of the interviews with patients and 77.14% (n = 27) of the interviews with clinicians were conducted by a female interviewer. Patients' interview duration ranged from 7.56 to 85.26 min, with an average length of 30.02 min (SD = 14.74). Interviews with clinicians were on average 30.97 min (SD = 12.43) long (range 10–70 min).

**Qualitative findings.** Overall, DIALOG+ was found to be acceptable to most interviewed clinicians and patients. Findings are presented per each TFA component construct and summarized in Table 4. S2 Table shows the intervention acceptability findings in more detail, including direct quotes from interviewees.

1. Affective Attitude

Experienced affective attitude is defined as how the clinicians and patients feel about DIALOG+, after taking part in the intervention. Clinicians and patients expressed very positive feelings about participating in the DIALOG+ sessions.

Table 3. Characteristics of interviewed participants, interviewers and methods used to conduct interviews.

| | Number (%) of patients interviewed (n = 40) | Number (%) clinicians interviewed (n = 35) |
|---|---|---|
| **Country** | | |
| Bosnia & Herzegovina | 8(20%) | 4 (11.43%) |
| Kosovo[a] | 8(20%) | 8(22.86%) |
| Montenegro | 7(17.5%) | 6(17.14%) |
| North Macedonia | 8(20%) | 8(22.86%) |
| Serbia | 9(22.5%) | 9(25.71%) |
| **Method used to conduct interviews** | | |
| Face-to-face | 11(27.5%) | 7 (20.00%) |
| Telephone call | 21(52.5%) | 26 (74.29%) |
| Video Call | 8(20%) | 2(5.71%) |
| **Professional background of interviewers** | | |
| Medical Doctor | 16(40%) | 12 (34.29%) |
| Psychiatrist | 9(22.5%) | 9 (25.71%) |
| Psychologist | 15(37.5%) | 14 (40%) |
| **Clinicians' professions** | | |
| Psychiatrist | – | 19 (54.29) |
| Nurse | – | 11 (31.43%) |
| Psychologist | – | 2 (5.71%) |
| Social Worker | – | 2 (5.71%) |
| Trainee | – | 1 (2.86%) |

[a]By United Nations Resolution.

Participating in DIALOG+ sessions felt positive for patients because it met their needs to be heard and empowered. Both patients and clinicians reported experiencing DIALOG+ sessions as enjoyable. Additionally, several participants found DIALOG+ sessions more relaxing in comparison to their usual treatment. However, participants' accounts also suggested that the intervention acceptability was reduced among some due to its perceived lack of flexibility and monotony. Some of the questions and domains covered by DIALOG+ were perceived as too intimate, and potentially limited the engagement of patients during the sessions.

2. Burden

The TFA component construct 'Burden' is concerned with the effort required to participate in the intervention. The intervention was perceived as complex and cognitively demanding for some, whereas for others it was perceived as simple to use and understand. Additionally, clinicians reported feeling disempowered to help with certain patients' life areas that they perceived as outside of their professional role, such as job search or any other material resources. Experienced feeling of disempowerment could have acted as a burden for clinicians during the intervention. Furthermore, the patients' booklet was perceived to require too much effort and was viewed as an obligation by some that may have impacted, in turn, their acceptability of the intervention. Overall, there was some cognitive and psychological burden experienced with participating in the DIALOG+ sessions.

3. Opportunity cost

Opportunity cost represents the benefits that had to be given up to engage with DIALOG+. Some participants reported additional workload associated with participating in the intervention, requiring them to skip their breaks and work longer

**Table 4.  Experienced acceptability-related themes per component constructs of the Theoretical Framework of Acceptability (TFA).**

| TFA components | THEMES | | | | |
|---|---|---|---|---|---|
| **1. AFFECTIVE ATTITUDE** | 1.1. Meeting patient's need for significance and empowerment | 1.2. Sessions perceived as enjoyable | 1.3. Sessions perceived as too rigid | 1.4. Conversations perceived as too personal | |
| **2. BURDEN** | 2.1. Variation in views about intervention complexity and required cognitive effort | 2.2. Clinicians reported feeling disempowered to help | 2.3. Using the patients' booklet required too much effort | | |
| **3. OPPORTUNITY-COST** | 3.1. Intervention perceived as additional workload | | | | |
| **4. ETHICALITY** | 4.1. Intervention perceived as a good fit to existing clinical practice | 4.2. Compatibility of intervention with perceived "best clinical practice" | | | |
| **5. SELF-EFFICACY** | 5.1. Ability to gradually develop confidence to receive and deliver the intervention | 5.2. Variable level of confidence to use the intervention | 5.3. Institutional support | | |
| **6. CLINICIANS' INTERVENTION COHERENCE** | 6.1. Understanding the main intervention principles | 6.2. Understanding the intervention's procedure | 6.3. Limited intervention coherence | 6.4. Tablet perceived as useful | 6.5. Patients' booklet perceived as useful |
| **7. PERCEIVED EFFECTIVENESS** | 7.1. Intervention perceived as effective | 7.2. Intervention effectiveness seen as dependent on illness and personal characteristics | 7.3. Intervention effectiveness is dependent on the degree of familiarity between clinicians and patients | 7.4. Limited experienced effectiveness | |

hours. However, the majority of participants did not report having to forgo any benefits as a result of their participating in DIALOG+, suggesting that the intervention largely showed high acceptability in relation to opportunity-costs.

4. Ethicality

Ethicality is the degree to which DIALOG+ is perceived to have good fit with the clinicians' and patients' value systems. DIALOG+ was perceived as similar to clinicians' regular work and as such also suitable for the existing value system. Many patients, however, indicated that DIALOG+ offers a more detailed approach in comparison with their usual care. This suggests that DIALOG+ intervention is a good fit with the current clinical practice. DIALOG+ was valued by both clinicians and patients because it fit with their view of "best clinical practice". DIALOG+ was seen to offer care during routine clinical meeting that was beyond medication adjustments, which was perceived as particularly valuable. Overall, participants reported that DIALOG+ was viewed as valuable.

5. Self-Efficacy

This component construct was defined as clinicians' and patients' confidence that they can participate in the DIALOG+ sessions. The ability of both clinicians and patients to allow themselves to gradually develop confidence to receive and deliver DIALOG+ emerged from the participants' accounts. This suggests that support and appropriate training programme can facilitate intervention acceptability. Additionally, it indicates the important role of time for the intervention deliverers and recipients to adapt to new practices, and eventually, adopt them. Participants' level of confidence in using the intervention depended on their previous familiarity with such therapeutic approaches and between the clinician-patient pair. Participants generally reported feeling confident about participating in DIALOG+ sessions. Clinicians did not perceive

any restrictions from their institutions to deliver the intervention, but they perceived that any institutional support received was more passive than active. Overall, clinicians and patients expressed moderate confidence that they can execute the behaviours needed to partake in the intervention.

6. Clinicians' Intervention Coherence

This component construct is about the extent to which the clinicians understood DIALOG+ and how it works to accomplish its benefits. Clinicians largely expressed a high level of coherence of the intervention's procedure and its main principles (e.g., patient-centeredness, patient involvement, positive reinforcement, solution-focused and structured communication, a more therapeutic relationship, opportunity for self-reflection and self-expression by patients) through which the intervention effects change, as intended. Clinicians' view was that the use of tablet facilitated conduct of DIALOG+ meetings and enabled them to economically organize, store, and use data and information relevant to patients. Clinicians also reported that they understood the usefulness of the patients' booklet during sessions. Accounts about the degree to which patients reported understanding DIALOG+ are included under intervention receipt in the results of the second study objective (subheading: Receipt of Intervention).

7. Perceived effectiveness

Experienced effectiveness is the degree to which DIALOG+ is perceived to have accomplished its main purpose. Participants experienced many of the intended benefits of DIALOG+, as originally envisaged by the trial designers. Clinicians and patients perceived DIALOG+ as highly likely to have improved patients' quality of life. Participants' views indicate that poor mental health prevented using the intervention as designed with some patients, which likely limited intervention acceptability. Certain personality characteristics, such as age, proactive attitude, open-mindedness and communicativeness were reported to influence intervention effectiveness. Already established familiarity between the clinician and patient when implementing a new intervention was perceived to act as both an enabler and a barrier to intervention effectiveness. Additionally, some participants did not observe any added value of using DIALOG+ compared to routine care and some felt that the number of intervention sessions delivered during the trial was insufficient to improve patients' condition.

Overall, the findings from the qualitative study suggest that mental health clinicians and patients with PSD, who were part of the intervention arm of the IMPULSE trial, found DIALOG+ to be moderately to highly acceptable. Using a multi-dimensional framework to investigate acceptability appeared to generate a depth and breadth of informative responses. In particular, findings from the acceptability study point to a number of ways in which acceptability of DIALOG+ could be optimized without compromising the fidelity of the intervention as originally designed. Potential strategies for optimizing DIALOG+ are presented below in the Discussion section.

## Intervention fidelity

Fidelity was investigated in terms of five domains: training, delivery, receipt, enactment (from [18]) and differentiation (from [19]). Findings relating to each domain are reported in the sections below. S3 Table presents the intervention fidelity findings in more detail.

**Training of clinicians.** All clinicians (n = 41, 100%) attended all DIALOG+ training sessions as intended. Additionally, researchers provided continuous support to clinicians during the trial either via individual contact/meetings or group meetings. This was done flexibly, with the opportunity to adjust according to individual needs; data relating to this support was not systematically collected. The initial training session was the longest – between 1 and 3 hours. The subsequent training sessions largely lasted up to 1 hour. They were organized both individually and as group training.

Additionally, researchers reported that all clinicians from the intervention arm completed all activities related to preparation and training for intervention delivery, such as receiving a tablet and practicing role-play vignettes. Only researchers from Bosnia and Herzegovina and Serbia reported that some of their clinicians did not read the manual thoroughly due to lack of time.

The clinicians' opinion of the training and supplementary materials was positive; they reported them to be useful in gaining confidence to deliver the novel intervention.

**Delivery of intervention.**

*Occurrence, attendance and duration of intervention sessions.* The proportion of DIALOG+ sessions that took place was 87.64% (n = 1,241) of the intended sessions. The majority of patients in the intervention arm (n = 189, 80.08%) attended all 6 sessions during the trial. 15 patients (6.36%) randomized to the intervention arm did not attend any sessions. The mean number of intervention sessions per patient was 5.26 (SD = 1.77). S4 Table contains information that summarizes the occurrence and attendance of all intervention sessions. Each DIALOG+ session was planned to last from 30 to 60 min. The duration of DIALOG+ sessions in the trial ranged from 4 to 90 min, with a mean duration of 28 min (SD = 11.69).

The most common method of delivery of intervention sessions was face-to-face (89.52%, n = 1,111). Few of the fifth and the majority of the sixth sessions were delivered remotely instead of in person due to pandemic restrictions.

After all first DIALOG+ sessions were completed, 32 clinicians (78.05%) reported not experiencing any problems with the delivery of DIALOG+ and participating in the study. The remaining clinicians (n = 9, 21.95%) reported problems related to technical difficulties and communication issues, the majority of which were recorded as successfully resolved.

*Life and treatment areas discussed in the four-step approach of the intervention.* In the interests of time efficiency, clinicians were instructed that no more than one domain should be selected for an in-depth discussion during the first DIALOG+ session and no more than three domains in second to sixth session. One domain was selected in 88.24% (n = 195) of the 1st DIALOG+ sessions as intended, and one to three domains were selected in 97.55% (n = 995) of the 2nd to 6th DIALOG+ sessions as planned. The mean number of domains selected per session was 1.8 (SD = 0.89).

The selection frequency of life and treatment areas during the intervention sessions is shown in Fig 1. The most frequently selected area to be discussed in the 4-step approach of the intervention was 'Mental Health' (28.93% of all delivered DIALOG+ sessions), whereas the least frequently selected area was 'Practical Help' (4.92% of all delivered DIALOG+ sessions).

*Actions agreed at the end of the intervention session.* The mean number of action items per DIALOG+ session was 2.52 (SD = 1.35). Most commonly, one to three action items were agreed during one DIALOG+ session

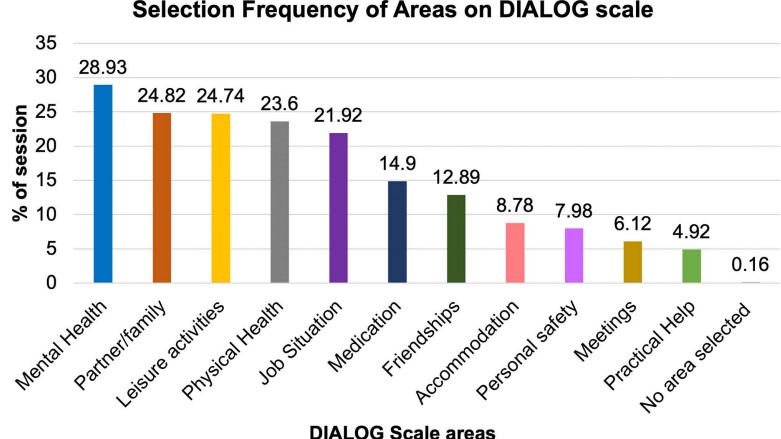

**Fig 1. A bar chart showing the selection frequency of life and treatment areas from the DIALOG scale (part of DIALOG+ intervention) in descending order for percentage of all delivered intervention sessions (n = 1,241).**

(84.21%). S3 Table contains more detailed information on frequency of action items agreed during the DIALOG+ sessions.

The qualitative data reported as action items were further analysed based on two pre-determined themes: a) person to whom the action is assigned; and b) type of action. When the qualitative data entered as one action was deemed to contain text that referred to more than one action, the analysis was done per each unique action. Reported data which was not considered an action was excluded in the analysis – 101 items were identified in the sample that did not contain an action item. Thus, after data cleaning, 3,126 unique action items set during the intervention sessions were identified. Table 5 contains a summary of the person who was responsible for a unique action and the type of action. A great majority of actions were assigned to the patient (82.15%, n = 2,651). Patient-led actions mostly related to connecting with others

Table 5. The frequency of action items per responsible person and type of action.

| Action items (responsible person/type of action) | (N) | % of Total |
|---|---|---|
| **Patient** | **2651** | **82.15%** |
| Connect with others | 488 | 15.12% |
| Healthier lifestyle | 468 | 14.50% |
| Attend/schedule doctor's appointment | 264 | 8.18% |
| Job/Education related activities | 248 | 7.69% |
| Speak or ask someone for help/support | 243 | 7.53% |
| Manage symptoms/Use coping strategies | 226 | 7.00% |
| Recreational activity | 194 | 6.01% |
| Domestic activity | 176 | 5.45% |
| Medication-related activities | 169 | 5.24% |
| Consider/Think of options | 67 | 2.08% |
| Seek information | 44 | 1.36% |
| Coping with the pandemic | 36 | 1.12% |
| Other | 28 | 0.87% |
| **Clinician** | **291** | **9.02%** |
| General support/advice and encouragement | 147 | 4.56% |
| Speak to family members/friends/colleagues | 44 | 1.36% |
| Medication review | 36 | 1.12% |
| Organize a home visit | 27 | 0.84% |
| Mare a referral | 17 | 0.53% |
| Seek information | 17 | 0.53% |
| Other | 3 | 0.09% |
| **Other (e.g., family member, friend)** | **178** | **5.52%** |
| General support/advice and encouragement | 151 | 4.68% |
| Accompany the patient | 17 | 0.53% |
| Help with taking medication regularly | 7 | 0.22% |
| Other | 3 | 0.09% |
| **Not an action** | **101** | **3.13%** |
| Not an action | 101 | 3.13% |
| **Unknown** | **6** | **0.19%** |
| Other | 4 | 0.12% |
| Manage symptoms/use coping strategies | 1 | 0.03% |
| Medication-related activities | 1 | 0.03% |
| **Total** | **3227** | **100%** |

and attempts to introduce healthier lifestyle habits (e.g., physical exercises, diets, sleep habits, etc.). It was not possible to determine the responsible person for 6 of the action items.

*Adherence to the intervention manual.* 37 audio recordings from the DIALOG+ sessions were obtained from 17/41 (41.46%) clinicians in the intervention arm. Our goal was to obtain one audio recording from each clinician in the intervention arm. This discrepancy occurred due to too few participants consenting to be audio recorded. Most recordings were of 3rd DIALOG+ sessions (n = 33), the rest were of 4th DIALOG+ sessions.

Each recording was scored against the 19 items of the DIALOG+ Adherence Scale [20]. More than one recording was collected from 6 clinicians, in which case an average score was calculated from each of the recordings. The DAS scores for the 17 clinicians in the intervention arm are shown in Table 6. Overall the mean adherence to the DIALOG+ manual was 13.88/19 (73.05%). Seven high-scoring items (mean score ≥ 0.90) were identified, and only one low-scoring item (mean score ≤ 0.25).

**Receipt of intervention.** We used Bellg et al.'s [18] definition of receipt to mean the patient side of fidelity, their engagement with and comprehension of the intervention sessions. The majority of patients in the audio recordings were

**Table 6. Descriptive statistics for DIALOG+ Adherence Scale (DAS) [20] scores for clinicians in the intervention and control arm and the number (%) of clinicians who delivered each of the DAS items.**

| DAS ITEMS | Intervention arm (n = 17 clinicians) | | | | | Control arm (n = 8 clinicians) | | |
|---|---|---|---|---|---|---|---|---|
| | Mean (SD) | Min | Max | Max. possible score | n(%) clinicians who delivered each DAS item | Mean (SD) | Min | Max |
| Reviewing actions | 0.57 (0.48) | 0 | 1 | 1 | 11 (64.76%) | 0.13 (0.35) | 0 | 1 |
| Satisfaction – DIALOG Scale | 0.99 (0.05) | 0.8 | 1 | 1 | 17 (100%) | 0 (0.00) | 0 | 0 |
| Review of ratings | 0.37 (0.45) | 0 | 1 | 1 | 8 (47.06%) | 0 (0.00) | 0 | 0 |
| Comparison | 0.09 (0.27) | 0 | 1 | 1 | 2 (11.76%) | 0 (0.00) | 0 | 0 |
| Positive reinforcement | 0.55 (0.49) | 0 | 1 | 1 | 10 (58.82%) | 0 (0.00) | 0 | 0 |
| Patient involvement in selecting areas | 0.82 (0.29) | 0 | 1 | 1 | 16 (94.12%) | 0.25 (0.46) | 0 | 1 |
| Number of areas | 0.99 (0.04) | 0.83 | 1 | 1 | 17 (100%) | 0.38 (0.52) | 0 | 1 |
| Step 1 – Understanding (explore) | 0.93 (0.24) | 0 | 1 | 1 | 16 (94.12%) | 0 (0.00) | 0 | 0 |
| Step 1 – Understanding (identify) | 0.78 (0.38) | 0 | 1 | 1 | 15 (88.24%) | 0.25 (0.46) | 0 | 1 |
| Step 2 – Looking forward (best case scenario) | 0.76 (0.40) | 0 | 1 | 1 | 14 (82.35%) | 0 (0.00) | 0 | 0 |
| Step 2 – Looking forward (small improvements) | 0.62 (0.45) | 0 | 1 | 1 | 12 (70.59%) | 0.25 (0.46) | 0 | 1 |
| Step 3 – Exploring options (patient) | 0.94 (0.18) | 0.4 | 1 | 1 | 17 (100%) | 0.38 (0.52) | 0 | 1 |
| Step 3 – Exploring options (clinician) | 0.84 (0.29) | 0 | 1 | 1 | 16 (94.12%) | 0.13 (0.35) | 0 | 1 |
| Step 3 – Exploring options (others) | 0.56 (0.49) | 0 | 1 | 1 | 11 (64.76%) | 0.13 (0.35) | 0 | 1 |
| Step 4 – Agreeing on actions | 0.97 (0.12) | 0.5 | 1 | 1 | 17 (100%) | 0.25 (0.46) | 0 | 1 |
| Step 4 – Agreeing on actions (summary) | 0.5 (0.46) | 0 | 1 | 1 | 11 (64.76%) | 0.13 (0.35) | 0 | 1 |
| Quality of interaction – positive regard | 0.9 (0.26) | 0 | 1 | 1 | 16 (94.12%) | 1 (0.00) | 1 | 1 |
| Quality of interaction – patient involvement in 4-step approach | 0.9 (0.25) | 0 | 1 | 1 | 16 (94.12%) | 0.13 (0.35) | 0 | 1 |
| Quality of interaction – clarity of actions | 0.87 (0.28) | 0 | 1 | 1 | 16 (94.12%) | 0.13 (0.35) | 0 | 1 |
| **DIALOG+ procedure (Select & Review)** | **4.38 (1.18)** | **2.5** | **7** | **7** | -- | **0.75 (1.04)** | **0** | **2** |
| **4-step procedure** | **6.9 (1.83)** | **3.6** | **9** | **9** | -- | **1.5 (2.27)** | **0** | **6** |
| **Quality of interaction** | **2.67 (0.44)** | **2** | **3** | **3** | -- | **1.25 (0.71)** | **1** | **3** |
| **Total score** | **13.88 (2.57)** | **8.8** | **17** | **19** | -- | **3.5 (3.55)** | **1** | **9** |

able to use the DIALOG scale, select at least one area for further discussion, discuss at least one area with the 4-step approach and suggest or agree actions to improve their satisfaction (Table 7).

Receipt was further explored during the end-of-trial qualitative interviews with patients and clinicians allocated to the intervention arm. Three intervention receipt-related themes were interpreted from the data through framework analysis (Table 8). The qualitative data regarding patients' receipt of the intervention suggest that they were largely able to recognize the steps involved in the intervention and how it works (e.g., patient-centeredness, patient involvement, therapeutic self-expression and self-reflection, a focus on solutions, comprehensive and structured communication), however patients' comprehension was perceived as limited among those with cognitive impairments.

**Enactment of agreed actions.** We defined enactment as patients attempting to complete, or completing, the action items agreed upon during the DIALOG+ session, in the target setting [18]. Table 9 shows the descriptive statistics of the enactment data collected from the audio recordings.

Enactment was further explored during the end-of-trial qualitative interviews with patients and clinicians allocated to the intervention arm. Table 10 presents the enactment-related themes that were interpreted from the data through framework analysis. The results suggest that many patients achieved the intended enactment of the agreed actions, however there were several external and internal factors, such as lack of support, financial resources, motivation, and poor mental health that hindered patients' enactment during the trial.

**Intervention differentiation.** Intervention differentiation refers to the difference between the intervention and the control condition, thus if the active intervention is indistinguishable from usual care, we would not expect the intervention to be effective. Besides data about occurrence, attendance and duration of control sessions, we also measured intervention differentiation by scoring audio recordings from control sessions against the DAS to investigate if and to what degree the content of the sessions matched the key elements of the intervention.

**Occurrence, attendance and duration of control sessions.** The proportion of control sessions that took place was 95.19% (n = 1,325) of the intended sessions. The majority of the patients in the control arm (n = 207, 89.22%) attended all

**Table 7. Patient level data on the receipt of the intervention as analysed from audio recordings from intervention sessions.**

|  | Number (%) of patients (N = 37) |
|---|---|
| Ability to rate satisfaction to all 11 areas of the DIALOG scale (Yes) | 36 (97.30) |
| Ability to select at least one area for discussion (Yes) | 35 (94.59) |
| At least one area discussed as part of the 4-step approach (Yes) | 36 (97.30) |
| Ability to suggest or agree actions (Yes) | 33 (89.19) |

**Table 8. Intervention receipt-related themes with direct quotes.**

| INTERVENTION RECEIPT-RELATED THEMES | |
|---|---|
| 1. Variable views about the level of patients' understanding of the intervention procedure | *"The difficulty was probably due to the nature of the patients' illness, that they probably didn't understand all the questions, so they gave the wrong answers because they misunderstood" (CKOS4)*<br>*"Patients understood the purpose, they understood the program and, they simply saw it as a good way to solve certain personal problems." (CMON3)* |
| 2. Comprehension of intended main intervention principles | *"In [DIALOG+] sessions we talk about the future, how to move forward." (PMON2)*<br>*"[with DIALOG+] I was able to go into more detail and I could start solving things." (PBOS8)*<br>*"[with DIALOG+] there is a certain method and order and everything is much easier then…" (PSER9)* |
| 3. Patients' booklet perceived as useful | *"I had that diary that we got, so I wrote down some things that were important to me - about myself, certain events, problems and so on. I also wrote down the doctor's advices there, and then I could occasionally remind myself." (PSER1)* |

**Table 9. Patient level data on the enactment of the agreed actions as analysed from audio recordings from intervention sessions.**

|  | Number (%) of patients (N = 37) |
|---|---|
| Attempt to complete at least one action (Yes) | 16 (43.24) |
| Completion of at least one action | 13 (35.14) |
| Ways patient accomplished the action: |  |
| a) By themselves/independently | 4 (10.81) |
| b) With the help of others | 6 (16.22) |
| c) Not specified | 3 (8.11) |

**Table 10. Enactment-related themes with direct quotes.**

| ENACTMENT-RELATED THEMES | |
|---|---|
| **1. Limited enactment of agreed actions** | |
| 1.1. Agreed actions were difficult to enact | *"I didn't complete a lot of homework tasks, I had a lot of problems with that"* (PSER8) |
| 1.2. External factors limited enactment | *"For example, I have this patient who loves history, loves sports, but his social circumstances are a disaster... so he barely has a pension, he doesn't have a computer...* (CSER8) |
| 1.3. Internal factors limited enactment | *"...but I did not always complete all of the task, I was simply in the phase where I could not do much."* (PBOS6)<br>*"It was rare for a patient to take the initiative to perform the assigned activities independently and responsibly."* (CMAC5) |
| **2. Enactment of agreed actions as intended** | |
| **2.1.** Agreed actions were enacted in daily life | *"I managed to complete all the agreed activities"* (PSER4)<br>*"We usually agreed on simple tasks, so they would generally complete them."* (CBOS3) |
| **2.2.** Well-tailored actions and a proactive attitude facilitated the implementation of actions in daily life | *"I was capable [to complete the actions] because I decided on some activities that were simple for me. And if I couldn't do an activity, I would tell [the clinician] that I can't, for example, to run."* (PKOS4)<br>*"I think DIALOG+ helped me the most when it comes to getting a job...we agreed on the task of printing those ads with phone numbers and posting them. I did that...it didn't take long and I got my first job."* (PSER9) |

6 sessions during the trial. Three patients (1.29%) from the control arm did not participate in any trial session. The mean number of sessions per patient was 5.71 (SD = 1.01). Table 11 summarizes the occurrence and attendance of all control sessions. The duration of routine clinical meetings in the participating countries was expected to range from 15–40 min. The duration of control sessions in the trial ranged from 2 to 60 min, with a mean duration of 20.14 min (SD = 11.22). The mean duration of both control and intervention sessions is shown in Fig 2. The mean difference between the duration of intervention and control sessions was 7.86 min (95% CI 6.97, 8.75; p < 0.001).

The most common method of delivery of control sessions was face-to-face (91.62%, n = 1,214). Some of the fifth and almost half of the sixth control sessions were provided remotely due to pandemic restrictions.

**Table 11. Control sessions' occurrence.**

| Number of control sessions | Session 1 | Session 2 | Session 3 | Session 4 | Session 5 | Session 6 | Total |
|---|---|---|---|---|---|---|---|
| **Intended** | 232 | 232 | 232 | 232 | 232 | 232 | **1392** |
| **Delivered** | 229 | 227 | 225 | 220 | 217 | 207 | **1325** |
| **% delivered/intended** | 98.71% | 97.84% | 96.98% | 94.83% | 93.53% | 89.22% | **95.19%** |
| **Number (%) of patients attending 1–6 sessions in total** | 2 (0.89) | 2 (0.89) | 5 (2.16%) | 3 (1.29%) | 10 (4.31) | 207 (89.22) | 232 (100) |

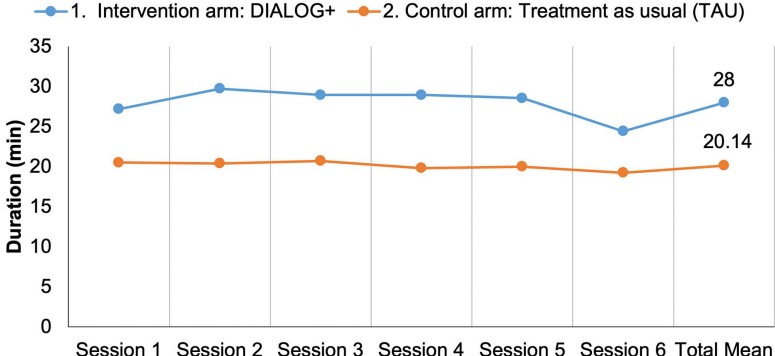

**Mean Duration of Trial Sessions**

**Fig 2. Mean duration of intervention and control sessions throughout the trial.**

**Adherence to the intervention manual and investigation of potential contamination in control group:** 16 audio recordings from control sessions were obtained from 8/40 (20%) clinicians in the control arm, significantly less than our goal of obtaining one audio recording from each clinician in the control arm (n = 40). This sample was smaller because too few participants consented to be audio recorded. The majority of the recordings (n = 13) were of 3rd control session, the rest were of 4th (n = 2) and 6th (n = 1) sessions.

Each recording was scored against the DAS [20]. More than one recording was collected from 2 clinicians, in which case an average score was calculated from each of the recordings. The DAS scores for the 8 clinicians in the control arm are shown in Table 6. The mean total score on the DAS was 3.5/19, compared with 13.88/19 for the recordings from the intervention sessions. There were also notable differences between the mean scores of the DAS subscales between the intervention and control sessions (S3 Table), further supporting the evidence of differentiation.

## Summary of process evaluation results

This process evaluation integrates many different data sources and yields an in-depth investigation of the IMPULSE cRCT and the DIALOG+ intervention as delivered and received during the trial. A number of contextual attributes in the participating countries, largely relating to the health services and systems, potentially negatively impacted on DIALOG+ acceptability, fidelity and outcomes. However, more attributes of context with the potential to have facilitated the intervention were identified. Of particular importance was the reported appetite for change of the existing mental health care among key stakeholders.

The experienced acceptability of the intervention was moderate to high. Participants felt positively about the intervention, perceived that a moderate amount of effort was required to take part in the intervention, reported little opportunity-costs associated with participating in the intervention, perceived DIALOG+ as a good fit with their value system, expressed moderate confidence that they can execute the behaviours needed to take part in the intervention, expressed a high level of understanding of the intervention and how it works and perceived DIALOG+ as highly likely to have achieved its purpose of improving patients' quality of life.

The fidelity of the intervention was high. The intervention training, delivery and attendance at intervention sessions were as planned. The duration of the sessions was slightly lower than envisaged. Regarding the content of the intervention sessions, the number of domains selected for discussion as well as the number of actions set occurred as planned. Most actions were assigned to the patient to complete before the next session, indicating high patient involvement during the sessions. A few problems with intervention delivery that were reported, such as technical issues and communication

**Table 12. Mapping of IMPULSE process evaluation findings to Omer et al.'s [5] mechanisms of actions of DIALOG+.**

| Omer et al.'s [5] proposed mechanisms of action of DIALOG+ | IMPULSE Process Evaluation Findings |
|---|---|
| A positive change in patients' specific areas of concern | At 6 months, the largest improvement of quality of life among patients in the intervention arm was observed in the areas 'friendships', 'living' and 'accommodation'. At 12 months, 'accommodation' no longer showed significant improvement [6]. These domains were not among the most commonly selected areas for discussion during the delivered DIALOG+ sessions (Fig 1). However, patients largely reported living with family members or partners [6], thus the improvements observed for 'living' could be due to the area 'partner/family' being the second most frequently discussed area during the intervention sessions. Additionally, the majority of actions agreed during the intervention related to connecting with others (Table 5), which could explain the improvements seen for the quality of life items 'friendship' and 'living' during the 6-month and 12-month outcome follow-ups. However, patients' enactment of actions was assessed as moderate. Our qualitative findings demonstrate that DIALOG+ enabled the clinical meetings to be focused on the main difficulties that the patient was experiencing and on finding crucial solutions to moving forward. |
| Comprehensive and solution-focused structure to the routine clinical meetings | The intervention acceptability and fidelity findings show that DIALOG+ enables an assessment of a comprehensive number of areas of a patient's life and treatment. All 11 domains included in DIALOG+ were chosen to be discussed at least once during the trial (Fig 1). Additionally, 2.52 action items were agreed on each DIALOG+ session, ensuring that the patient is left with concrete steps that address patients' needs (Table 5). Participants reported that the structure of DIALOG+ enabled patients to improve areas of their life and treatment that there were dissatisfied with, ensured that the communication with their clinician was comprehensive, which in turn was perceived to deepen the clinician-patient relationship, and made it easier for people to participate in the intervention. Participants' view was that DIALOG+ broadened the focus of clinical discussions beyond pharmacotherapy, which was seen as particularly valuable among both clinicians and patients. This is supported by the types of action items agreed during the DIALOG+ sessions (Table 5). Additionally, the tablet was perceived as an important facilitator to achieving a comprehensive and solution-focused structure during DIALOG+ sessions. The patients' booklet was also seen to serve as a good reminder of the agreed actions between sessions that kept patients organized, however some patients found it challenging to use. |
| Opportunity for self-reflection and therapeutic self-expression | The initial assessment done as part of the intervention prompted patients to reflect on 11 life and treatment areas, which enabled a greater understanding of the areas where the patient was experiencing difficulties or areas where the patient was doing well. Participants reported that DIALOG+ provided more time and space for patients to express their feelings, needs and concerns, thus it increased their understanding of themselves and their condition. This is supported by the DIALOG+ Adherence Scale scores. The identification of positive areas that the patient was satisfied with, was seen as having a therapeutic effect. DIALOG+-led conversations were perceived to help strengthen the therapeutic relationship and improve the quality of communication. Clinicians also reported gaining a greater awareness of the patients' overall condition. Some patients reported using the booklet as a tool that facilitated self-reflection and self-expression. |
| Empowerment | With DIALOG+, patients were empowered to take control. The content analysis of the reported action items showed that a large majority of agreed actions were the responsibility of the patients (82.15%). Additionally, participants reported active patient involvement during the DIALOG+ sessions and patients taking more initiative in their treatment. This is supported by the DIALOG+ Adherence Scale scores. The intervention was also perceived to have met patients' needs to be heard and empowered. |

concerns, were successfully resolved. Additionally, the adherence to the DIALOG+ manual was moderate. Patients' receipt of the intervention was high, with certain limitations among patients perceived to have cognitive impairments. Patients' enactment of actions was moderate. Factors that appear to have limited enactment include lack of motivation and poor mental health, as well as lack of access to resources. The delivery, attendance and duration of the control sessions were also as intended and no contamination between intervention and control groups was identified. Thus, intervention differentiation was also high.

In regard to the differences between the DIALOG+ intervention and treatment as usual during routine clinical meetings, DIALOG+ was perceived to foster a more relaxed atmosphere and to offer opportunity for the patients to express themselves more and understand themselves better. They were perceived to enable more comprehensive and structured conversations between the patients and clinicians and thus to establish a stronger

clinician-patient relationship. DIALOG+ was seen to encourage more patient involvement, and to ensure a solution-focused approach to care, while reducing the emphasis solely on medication. The intervention also utilized tablet computers during clinical meetings. Routine clinical meetings are usually not led by nurses, which was the case with several clinicians delivering DIALOG+. Additionally, DIALOG+ sessions lasted longer, and were perceived by some to be more monotonous, too intimate and more complex. Importantly, some clinicians did not observe any difference between usual treatment and DIALOG+ . However, upon analysis of audiotapes from control and intervention sessions, it was concluded that treatment as usual was meaningfully distinct from DIALOG+ .

To further improve our understanding about how DIALOG+ works, the integration of the findings was mapped to Omer et al.'s [5] proposed mechanisms of actions of DIALOG+ in Table 12.

## Discussion

This process evaluation brings together findings from both quantitative and qualitative methods providing a rich, contextualized understanding of clinicians' and patients' engagement with DIALOG+ and its proposed mechanisms of action. There were several contextual attributes regarding patients, clinicians and mental health services that might have impaired DIALOG+ acceptability, fidelity and outcomes, such as resource limitations, funding priorities, reliance on paper records and lack of community support. Nevertheless, the study also identified several attributes of context that might have acted as facilitators to engagement with the intervention, such as appetite for change among key stakeholders. Acceptability of the studied intervention was moderate to high and its fidelity was high.

The results from this process evaluation, in particular, the fidelity findings, indicate that the findings from the IMPULSE trial represent a valid assessment of the intervention as designed. The observed moderate to high intervention acceptability is likely to have contributed to high intervention fidelity by clinicians and patients during the trial, which in turn contributed to the positive effect of the intervention on patients' quality of life. An alternative explanation to the trial findings that should also be considered includes the longer amount of time that patients in the intervention arm spent with their treating clinician (+7.86 min), which may have partly contributed in itself towards better patient outcomes. The identified contextual barriers appear not to have impaired the intervention fidelity and acceptability of the trial participants. This is likely due to the high level of resources that were available during the trial compared to usual practice. Thus, the identification of the contextual barriers should not mean that DIALOG+ is not feasible in the contexts explored. These barriers, however, may make future implementation and scaling-up of DIALOG+, as well as community mental health services, more challenging. Making efforts to overcome these barriers are not only beneficial for DIALOG+ implementation, but also for overall improvement of community mental health services in this region. Importantly, we identified a large appetite for change among the clinicians, patients, policymakers and carers regarding the current mental health care in the participating countries.

In comparison with the process evaluation of the UK DIALOG+ trial [5], the mean number of action items per DIALOG+ session in the IMPULSE trial was lower (4.9 vs 2.52, respectively). In the IMPULSE trial, patients were responsible for a higher proportion of actions. Actions related to healthier lifestyle were among the most common types of actions in both trials. However, the majority of actions for patients in the IMPULSE trial were related to connecting with others, unlike in the UK trial where engagement with treatment, or raising issues with healthcare professionals were among the most common. Most clinician-led actions were related to general support, whereas in the UK trial they were related to providing practical assistance. The most commonly discussed domains in the IMPULSE trial were mental health, partner/family, leisure activities and physical health, whereas in the UK trial they were mental health, physical health, and accommodation. The mean number of domains selected in the IMPULSE trial (1.8, SD = 0.89) was lower than in the UK trial (2.3, SD = 1.1). The observed variations between the UK and IMPULSE trial are likely related to cultural differences and differences in the settings where the trials took place.

Our findings support Omer et al.'s [5] suggestion that the effect of DIALOG+ on patients' quality of life is partly achieved via structured discussions of specific areas of concern followed by triggering positive change by agreeing on actions related to those areas. Similarly to Omer et al.'s findings [5], patients' physical health was also frequently addressed during the DIALOG+ sessions and the second most commonly agreed action item was related to healthier lifestyle, yet DIALOG+ did not show any significant effect on this domain at 6 and 12 months. This could be explained by the particular obstacles that this patient population face in bettering such outcomes [26]. Our findings are in line with the remaining proposed mechanisms that DIALOG+ works through establishing a comprehensive and solution-focused structure to routine mental health clinical meetings, providing an opportunity for patient's self-reflection and self-expression, and patient empowerment. The shift away from discussions primarily focused on patients' medication during routine clinical meetings and the establishment of a stronger therapeutic relationship through DIALOG+ were particularly highlighted in our context of Southeast Europe, whereas this was not the case in the UK study [5]. Additionally, the booklet that patients received to use for writing down the agreed actions during DIALOG+ sessions was perceived to facilitate the comprehensive and solution-focused structure of DIALOG+, as well as patients' self-reflection and expression.

## Strengths and limitations

To the best of our knowledge, this is the first process evaluation of a non-pharmacological cRCT with people with PSD in Southeast Europe, and one of few such efforts globally. It is not only a comprehensive exploration of the mental health care in this region, which has been referred as "a blind spot on the global mental health map" [27], but also of implementation and potential sustainability of a non-pharmacological intervention overall. The multi-method approach of this process evaluation and its use of published frameworks to explore the role of context, acceptability of the intervention and the fidelity are major strengths of this study. By integrating the different data sources and data types, this study yields a thorough knowledge which is 'greater than the sum of the parts', generated from all five participating LMICs in Southeast Europe. Additionally, all data was collected prior to researchers' or participants' awareness of the trial outcomes, which minimized data collection bias. Furthermore, all researchers who participated in the qualitative study received an in-depth training programme in qualitative methods, which reduced the risk of varying levels of quality and depth of information collected across the different interviews. The qualitative data was coded in the original languages and researchers from all participating countries collaborated in the framework analysis process. This minimized the loss of contextual-specific meaning and traits.

This study also has several limitations. The number of audio recordings obtained from the trial sessions was fewer than the anticipated one recording per each clinician, adding a risk of sampling bias. The duration of control sessions and some intervention sessions was collected from clinicians, making it subject to clinicians' memory and social desirability bias. Moreover, only a subgroup of all trial participants were interviewed at the end of the trial, which may have introduced selection bias.

Strengths and limitations of the IMPULSE cRCT are reported in detail elsewhere [6].

## Future implications

The current study findings can inform strategies for further scaling and sustainability of a digital, psychosocial intervention such as DIALOG+. The DIALOG+ intervention may not be well-suited for all PSD patients or during every routine clinical meeting and may need tailoring to different clinical situations. The findings from this study suggest that it could be most suitable for outpatients open to communication, reflection and change. The intervention seems to be more acceptable to younger patients and clinicians. The frequency and intervals between sessions should be flexible, based on the patient's needs. Further monitoring of intervention delivery and patients' enactment of agreed actions, as well as availability of continuous supervision for clinicians, could support healthcare systems to track long-term implementation of DIALOG+, and potential lasting effects on patients and clinicians.

Future DIALOG+ training should particularly focus on intervention steps related to the initial DIALOG scale and review of patients' ratings, as these had the lowest adherence. Additionally, the training should provide more guidance about actions set at the end of the intervention session. To improve the enactment of actions, they should be well-tailored to the individual patient's needs and abilities. Based on our findings (Tables 6 and 11), we suggest the use of patients' booklets could be optional for the patient.

The mean difference in length of control and intervention sessions could inform future evaluations of DIALOG+ that should consider a comparison group where the clinical meetings would last similar to the average duration of DIALOG+ sessions to account for the possible "attention effect" – a longer amount of time spent with treating clinician. Future research should also explore evidence-based techniques for addressing the identified contextual barriers in low-resource mental health care settings.

## Conclusions

Process evaluations of complex health interventions nested within trials are crucial to understanding what was delivered, how, and what mechanisms of action may be at play, so that the interventions can be replicated in practice. The moderate to high intervention acceptability is likely to have contributed to the observed high intervention fidelity by clinicians and patients, which in turn contributed to the positive effect of the intervention on patients' quality of life. DIALOG+ as delivered and received closely resembled that which was planned by the trial designers in accordance with the data-driven and theory-informed implementation strategy. The control sessions were substantially distinct from the intervention. The identified contextual barriers appear not to have impaired intervention acceptability and fidelity but could pose challenges for future intervention sustainability. The study has implications for overall implementation and sustainability of DIALOG+ and other similar mental health interventions in low- and middle-income countries.

## Supporting information

**S1 File. Interview topic guides with clinicians and patients from the intervention arm of the IMPULSE trial.** List of questions asked during interviews with patients and clinicians.
(DOCX)

**S1 Table. Findings from data analysis and triangulation of pre-trial focus groups (General FGs), initial site visits and mental health policy analysis per each contextual attribute from Squires et al. (2019).** Summary of results, divided into barriers, facilitators, and triangulation, from three data sources per each contextual attribute.
(DOCX)

**S2 Table. Intervention acceptability-related themes with direct quotes per component construct of the Theoretical Framework of Acceptability.** In-depth description of all themes interpreted from the interviews with clinicians and patients with illustrative quotes.
(DOCX)

**S3 Table. Intervention fidelity findings related to training, delivery, receipt, enactment and differentiation.** Additional in-depth findings related to each investigated domain of intervention fidelity.
(DOCX)

## Acknowledgments

We would like to thank all the study participants who gave their valuable time to take part in this study. We also thank the wider IMPULSE study team, without whose contributions this study would not have been possible.

## Author contributions

**Conceptualization:** Tamara Pemovska, Nikolina Jovanovic, Tamara Radojičić, Jill J Francis.

**Data curation:** Tamara Pemovska.

**Formal analysis:** Tamara Pemovska, Tamara Radojičić, Silvana Markovska Simoska, Fjolla Ramadani, Sanja Andrić Petrović, Emina Karamehić, Biljana Blazhevska Stoilkovska, Jon Konjufca.

**Funding acquisition:** Nikolina Jovanovic, Alma Džubur Kulenović, Lidija Injac Stevović, Jill J Francis.

**Investigation:** Tamara Radojičić, Silvana Markovska Simoska, Fjolla Ramadani, Sanja Andrić Petrović, Emina Karamehić, Biljana Blazhevska Stoilkovska, Jon Konjufca, Stefan Jerotić.

**Methodology:** Tamara Pemovska, Nikolina Jovanovic, Tamara Radojičić, Jill J Francis.

**Project administration:** Tamara Pemovska.

**Resources:** Tamara Pemovska, Nikolina Jovanovic.

**Supervision:** Nikolina Jovanovic, Alma Džubur Kulenović, Lidija Injac Stevović, Jill J Francis.

**Validation:** Tamara Pemovska, Tamara Radojičić, Silvana Markovska Simoska, Fjolla Ramadani, Sanja Andrić Petrović, Emina Karamehić, Biljana Blazhevska Stoilkovska, Jon Konjufca.

**Visualization:** Tamara Pemovska.

**Writing – original draft:** Tamara Pemovska, Nikolina Jovanovic, Tamara Radojičić, Jill J Francis.

**Writing – review & editing:** Tamara Pemovska, Nikolina Jovanovic, Tamara Radojičić, Silvana Markovska Simoska, Fjolla Ramadani, Sanja Andrić Petrović, Emina Karamehić, Biljana Blazhevska Stoilkovska, Jon Konjufca, Stefan Jerotić, Alma Džubur Kulenović, Lidija Injac Stevović, Jill J Francis.

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
