## [Decision Letter · Decision Letter 0]

6 Jun 2025

Dear Dr. Jovanovic,

Thank you for submitting your manuscript to PLOS ONE. After careful consideration, we feel that it has merit but does not fully meet PLOS ONE’s publication criteria as it currently stands. Therefore, we invite you to submit a revised version of the manuscript that addresses the points raised during the review process.

We look forward to receiving your revised manuscript.

Kind regards,

Cheong Kim

Academic Editor

PLOS ONE

2. Please note that in order to use the direct billing option the corresponding author must be affiliated with the chosen institute. Please either amend your manuscript to change the affiliation or corresponding author, or email us at plosone@plos.org with a request to remove this option.

3. In the online submission form you indicate that your data is not available for proprietary reasons and have provided a contact point for accessing this data. Please note that your current contact point is a co-author on this manuscript. According to our Data Policy, the contact point must not be an author on the manuscript and must be an institutional contact, ideally not an individual. Please revise your data statement to a non-author institutional point of contact, such as a data access or ethics committee, and send this to us via return email. Please also include contact information for the third party organization, and please include the full citation of where the data can be found.

Reviewers' comments:

Reviewer's Responses to Questions

**Comments to the Author**

1. Is the manuscript technically sound, and do the data support the conclusions?

Reviewer #1: Yes

Reviewer #2: Yes

2. Has the statistical analysis been performed appropriately and rigorously?

Reviewer #1: Yes

Reviewer #2: Yes

3. Have the authors made all data underlying the findings in their manuscript fully available?

Reviewer #1: Yes

Reviewer #2: Yes

4. Is the manuscript presented in an intelligible fashion and written in standard English?

Reviewer #1: Yes

Reviewer #2: Yes

Reviewer #1: Scientific and social relevance: addresses psychosocial interventions in resource-limited contexts.

Robust methodology: use of mixed methods with qualitative and quantitative data collection, triangulation of sources, and use of recognized theoretical frameworks.

Clarity in presentation: logically structured and well-organized, with tables and detailed descriptions.

Suggestion for additional clarity: although it already includes many details, some tables could be moved to supplementary material to improve the flow of the main text.

Technical terms: ensure that all less familiar terms, such as “intervention fidelity” and “solution-focused approach,” are briefly explained within the body of the text for readers less familiar with them.

Reviewer #2: A clearer match between the stated aims and the main findings would improve the text. Although acceptance, faithfulness, and contextual influence are described in the study, the discussion should further explain how these connect to the primary IMPULSE trial's observed clinical outcomes.

Purposive sampling is acceptable, but more information is required to understand how site variety and demographic diversity were maintained. Transparency could be improved by including a table that summarizes participant characteristics by site.

**Do you want your identity to be public for this peer review?** For information about this choice, including consent withdrawal, please see our Privacy Policy

Reviewer #1: No

Reviewer #2: No

---

## [Author Response · Author response to Decision Letter 1]

27 Jul 2025

Response to reviewers: We have revised the manuscript where needed to align with PLOS ONE’s style requirements and those for file naming, using the templates provided.

2. Please note that in order to use the direct billing option the corresponding author must be affiliated with the chosen institute. Please either amend your manuscript to change the affiliation or corresponding author, or email us at plosone@plos.org with a request to remove this option.

Response to reviewers: The correct affiliation of the corresponding author is as noted in the manuscript ‘Centre for Psychiatry and Mental Health, Wolfson Institute of Population Health, Queen Mary University of London, London, United Kingdom’

3. In the online submission form you indicate that your data is not available for proprietary reasons and have provided a contact point for accessing this data. Please note that your current contact point is a co-author on this manuscript. According to our Data Policy, the contact point must not be an author on the manuscript and must be an institutional contact, ideally not an individual. Please revise your data statement to a non-author institutional point of contact, such as a data access or ethics committee, and send this to us via return email.

Response to reviewers: The dataset (which includes individual interview transcripts) used and analysed during the current study is not publicly available due to participant confidentiality and ethical considerations. The authors do not have the ethical approval to make data immediately available to the public. The study did not include explicit consent for public data sharing. The data are part of the IMPULSE trial, whose chief investigator was Dr Nikolina Jovanović (the co-author previously listed as the contact point regarding data). We have revised the data statement to a non-author institutional contact in line with PLOS ONE’s data policy, and data access requests may be directed to Queen Mary University of London, Wolfson Institute of Population Health; email: wiph-admin@qmul.ac.uk.

Response to reviewers: The ethics statement appears only in the Methods section of the manuscript.

Reviewers' comments:

Reviewer's Responses to Questions

Comments to the Author

1. Is the manuscript technically sound, and do the data support the conclusions? The manuscript must describe a technically sound piece of scientific research with data that supports the conclusions. Experiments must have been conducted rigorously, with appropriate controls, replication, and sample sizes. The conclusions must be drawn appropriately based on the data presented.

Reviewer #1: Yes

Reviewer #2: Yes

Response to reviewers: We thank both reviewers for affirming that the manuscript is technically sound and that the data support the conclusions.

2. Has the statistical analysis been performed appropriately and rigorously?

Reviewer #1: Yes

Reviewer #2: Yes

Response to reviewers: We appreciate the reviewers’ positive assessment of the statistical analyses conducted.

3. Have the authors made all data underlying the findings in their manuscript fully available?

Reviewer #1: Yes

Reviewer #2: Yes

Response to reviewers: Thank you. As noted above, we have amended the Data Availability Statement to better reflect PLOS ONE’s policies and to provide a non-author institutional contact.

4. Is the manuscript presented in an intelligible fashion and written in standard English?

Reviewer #1: Yes

Reviewer #2: Yes

Response to reviewers: We are grateful for this positive feedback.

5. Review Comments to the Author

Reviewer #1: Scientific and social relevance: addresses psychosocial interventions in resource-limited contexts. Robust methodology: use of mixed methods with qualitative and quantitative data collection, triangulation of sources, and use of recognized theoretical frameworks. Clarity in presentation: logically structured and well-organized, with tables and detailed descriptions.

Suggestion for additional clarity: although it already includes many details, some tables could be moved to supplementary material to improve the flow of the main text.

Technical terms: ensure that all less familiar terms, such as “intervention fidelity” and “solution-focused approach,” are briefly explained within the body of the text for readers less familiar with them.

Response to reviewers: Thank you for the positive feedback and suggestions. We have addressed them as follows:

• Moved Table 5 (occurrence and attendance of all intervention sessions) and Table 6 (frequency of action items agreed during the DIALOG+ sessions) to the existing supplementary material related to intervention fidelity findings (S4 Table) to improve the flow of the main text.

• The term “intervention fidelity” is already briefly explained in the Abstract and Introduction. We have now also added a short clarification for “solution-focused approach” at first mention in the Introduction and Methods section to aid reader understanding (see pages 5 and 7 in the revised manuscript, respectively).

Reviewer #2: A clearer match between the stated aims and the main findings would improve the text. Although acceptance, faithfulness, and contextual influence are described in the study, the discussion should further explain how these connect to the primary IMPULSE trial's observed clinical outcomes.

Purposive sampling is acceptable, but more information is required to understand how site variety and demographic diversity were maintained. Transparency could be improved by including a table that summarizes participant characteristics by site.

Response to reviewers: We appreciate these thoughtful comments and have addressed them as follows:

• In the Discussion section, we have included how the process evaluation findings regarding acceptability, fidelity and the role of context relate to the IMPULSE trial’s observed effect of the intervention on patients’ quality of life (see page 36).

• The manuscript already includes details on the purposive sampling approach, specifying efforts to ensure variation in participants for interviews regarding intervention acceptability (page 9). To further improve transparency, we have now included additional information in the supplementary material (Table S3), which presents characteristics of interviewed patients (age, gender, diagnosis, level of engagement with DIALOG+) by site. Associated edits in the manuscript are made on page 19.

6. PLOS authors have the option to publish the peer review history of their article (what does this mean?). If published, this will include your full peer review and any attached files. Do you want your identity to be public for this peer review? For information about thischoice, including consent withdrawal, please see our Privacy Policy.

Reviewer #1: No

Reviewer #2: No

Response to reviewers: Noted - we respect and acknowledge the reviewers' preferences.

---

## [Decision Letter · Decision Letter 1]

24 Nov 2025

Process evaluation of a hybrid effectiveness-implementation, pragmatic, cluster randomised controlled trial (IMPULSE) to improve psychosocial treatment of patients with psychotic-spectrum disorders in Southeast Europe

PONE-D-25-20318R1

Dear Dr. Jovanovic,

We’re pleased to inform you that your manuscript has been judged scientifically suitable for publication and will be formally accepted for publication once it meets all outstanding technical requirements.

Kind regards,

Cheong Kim

Academic Editor

PLOS ONE

Additional Editor Comments (optional):

Reviewers' comments:

Reviewer's Responses to Questions

**Comments to the Author**

Reviewer #1: All comments have been addressed

Reviewer #3: All comments have been addressed

2. Is the manuscript technically sound, and do the data support the conclusions?

Reviewer #1: Yes

Reviewer #3: Yes

3. Has the statistical analysis been performed appropriately and rigorously?

Reviewer #1: Yes

Reviewer #3: Yes

4. Have the authors made all data underlying the findings in their manuscript fully available?

Reviewer #1: Yes

Reviewer #3: Yes

5. Is the manuscript presented in an intelligible fashion and written in standard English?

Reviewer #1: Yes

Reviewer #3: Yes

Reviewer #1: The revised manuscript fully meets the journal's requirements and satisfactorily addresses the reviewers' comments. Its methodological clarity, relevance of the topic, and rigor in execution make it a valuable contribution to the field of mental health in resource-limited settings.

Reviewer #3: The manuscript provides an insightful and rigorously conducted process evaluation of the IMPULSE hybrid effectiveness-implementation, pragmatic cluster RCT aiming to improve psychosocial treatment for individuals with psychotic-spectrum disorders across Southeast Europe. The topic is of high relevance, and the study meaningfully addresses a well-recognized gap in mental health implementation research in low- and middle-resource settings.

Strengths include:

A clear theoretical underpinning using established implementation frameworks.

Strong integration of mixed-methods data, with transparent reporting of analytical procedures.

Thoughtful examination of contextual influences and implementation variability across countries.

Clear articulation of how implementation outcomes informed interpretation of the trial’s effectiveness results.

High overall coherence, readability, and adherence to PLOS ONE reporting expectations.

**Do you want your identity to be public for this peer review?** For information about this choice, including consent withdrawal, please see our Privacy Policy

Reviewer #1: No

Reviewer #3: No

---

## [Editor Report · Acceptance letter]

PONE-D-25-20318R1

PLOS One

Dear Dr. Jovanovic,

I'm pleased to inform you that your manuscript has been deemed suitable for publication in PLOS One. Congratulations! Your manuscript is now being handed over to our production team.

Kind regards,

on behalf of

Dr. Cheong Kim

Academic Editor

PLOS One